# Does Unsupervised Architecture Representation Learning Help Neural Architecture Search?

**Shen Yan, Yu Zheng, Wei Ao, Xiao Zeng, Mi Zhang**
Michigan State University
{yanshen6,zhengy30,aowei,zengxia6,mizhang}@msu.edu

## Abstract

Existing Neural Architecture Search (NAS) methods either encode neural architectures using discrete encodings that do not scale well, or adopt supervised learning-based methods to jointly learn architecture representations and optimize architecture search on such representations which incurs search bias. Despite the widespread use, architecture representations learned in NAS are still poorly understood. We observe that the structural properties of neural architectures are hard to preserve in the latent space if architecture representation learning and search are coupled, resulting in less effective search performance. In this work, we find empirically that pre-training architecture representations using only neural architectures without their accuracies as labels improves the downstream architecture search efficiency. To explain this finding, we visualize how unsupervised architecture representation learning better encourages neural architectures with similar connections and operators to cluster together. This helps map neural architectures with similar performance to the same regions in the latent space and makes the transition of architectures in the latent space relatively smooth, which considerably benefits diverse downstream search strategies.

## 1 Introduction

Unsupervised representation learning has been successfully used in a wide range of domains including natural language processing [1, 2, 3], computer vision [4, 5], robotic learning [6, 7], and network analysis [8, 9]. Although differing in specific data type, the root cause of such success shared across domains is learning good data representations that are independent of the specific downstream task. In this work, we investigate unsupervised representation learning in the domain of neural architecture search (NAS), and demonstrate how NAS search spaces encoded through unsupervised representation learning could benefit the downstream search strategies.

Standard NAS methods encode the search space with the adjacency matrix and focus on designing different downstream search strategies based on reinforcement learning [10], evolutionary algorithm [11], and Bayesian optimization [12] to perform architecture search in discrete search spaces. Such discrete encoding scheme is a natural choice since neural architectures are discrete. However, the size of the adjacency matrix grows quadratically as search space scales up, making downstream architecture search less efficient in large search spaces [13]. To reduce the search cost, recent NAS methods employ dedicated networks to learn continuous representations of neural architectures and perform architecture search in continuous search spaces [14, 15, 16, 17]. In these methods, architecture representations and downstream search strategies are jointly optimized in a supervised manner, guided by the accuracies of architectures selected by the search strategies. However, these supervised architecture representation learning-based methods are biased towards weight-free operations (e.g., skip connections, max-pooling) which are often preferred in the early stage of the search process, resulting in lower final accuracies [18, 19, 20, 21].

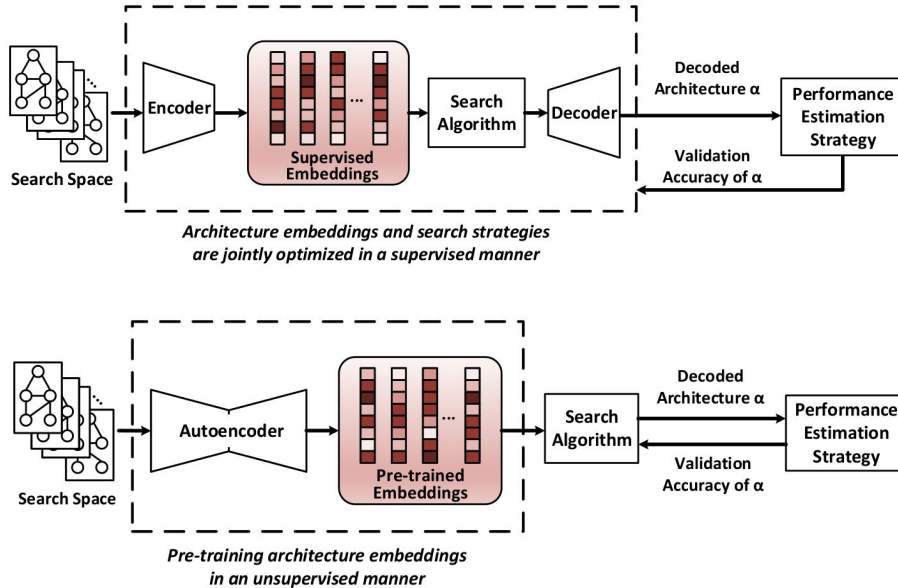

Figure 1: Supervised architecture representation learning (top): the supervision signal for representation learning comes from the accuracies of architectures selected by the search strategies. *arch2vec* (bottom): disentangling architecture representation learning and architecture search through unsupervised pre-training.

In this work, we propose *arch2vec*, a simple yet effective neural architecture search method based on unsupervised architecture representation learning. As illustrated in Figure 1, compared to supervised architecture representation learning-based methods, *arch2vec* circumvents the bias caused by joint optimization through decoupling architecture representation learning and architecture search into two separate processes. To achieve this, *arch2vec* uses a variational graph isomorphism autoencoder to learn architecture representations using only neural architectures without their accuracies. As such, it injectively captures the local structural information of neural architectures and makes architectures with similar structures (measured by edit distance) cluster better and distribute more smoothly in the latent space, which facilitates the downstream architecture search. We visualize the learned architecture representations in §4.1. It shows that architecture representations learned by *arch2vec* can better preserve structural similarity of local neighborhoods than its supervised architecture representation learning counterpart. In particular, it is able to capture topology (e.g. skip connections or straight networks) and operation similarity, which helps cluster architectures with similar accuracy.

We follow the NAS best practices checklist [22] to conduct our experiments. We validate the performance of *arch2vec* on three commonly used NAS search spaces NAS-Bench-101 [23], NAS-Bench-201 [24] and DARTS [15] and two search strategies based on reinforcement learning (RL) and Bayesian optimization (BO). Our results show that, with the same downstream search strategy, *arch2vec* consistently outperforms its discrete encoding and supervised architecture representation learning counterparts across all three search spaces.

Our contributions are summarized as follows:

- We propose a neural architecture search method based on unsupervised representation learning that decouples architecture representation learning and architecture search.

- We show that compared to supervised architecture representation learning, pre-training architecture representations without using their accuracies is able to better preserve the local structure relationship of neural architectures and helps construct a smoother latent space.

- The pre-trained architecture embeddings considerably benefit the downstream architecture search in terms of efficiency and robustness. This finding is consistent across three search spaces, two search strategies and two datasets, demonstrating the importance of unsupervised architecture representation learning for neural architecture search.

The implementation of *arch2vec* is available at https://github.com/MSU-MLSys-Lab/arch2vec.

## 2 Related Work

**Unsupervised Representation Learning of Graphs**. Our work is closely related to unsupervised representation learning of graphs. In this domain, some methods have been proposed to learn representations using local random walk statistics and matrix factorization-based learning objectives [8, 9, 25, 26]; some methods either reconstruct a graph's adjacency matrix by predicting edge existence [27, 28] or maximize the mutual information between local node representations and a pooled graph representation [29]. The expressiveness of Graph Neural Networks (GNNs) is studied in [30] in terms of their ability to distinguish any two graphs. It also introduces Graph Isomorphism Networks (GINs), which is proved to be as powerful as the Weisfeiler-Lehman test [31] for graph isomorphism. [32] proposes an asynchronous message passing scheme to encode DAG computations using RNNs. In contrast, we injectively encode architecture structures using GINs, and we show a strong pre-training performance based on its highly expressive aggregation scheme. [33] focuses on network generators that output relational graphs, and the predictive performance highly depends on the structure measures of the relational graphs. In contrast, we encode structural information of neural networks into compact continuous embeddings, and the predictive performance depends on how well the structure is injected into the embeddings.

**Regularized Autoencoders**. Autoencoders can be seen as energy-based models trained with reconstruction energy [34]. Our goal is to encode neural architectures with similar performance into the same regions of the latent space, and to make the transition of architectures in the latent space relatively smooth. To prevent degenerated mapping where latent space is free of any structure, there is a rich literature on restricting the low-energy area for data points on the manifold [35, 36, 37, 38, 39]. Here we adopt the popular variational autoencoder framework [37, 27] to optimize the variational lower bound w.r.t. the variational parameters, which as we show in our experiments acts as an effective regularization. While [40, 41] use graph VAE for the generative problems, we focus on mapping the finite discrete neural architectures into the continuous latent space regularized by KL-divergence such that each architecture is encoded into a unique area in the latent space.

**Neural Architecture Search (NAS)**. As mentioned in §1, early NAS methods are built upon discrete encodings [42, 43, 12, 44, 45], which face the scalability challenge [46, 47] in large search spaces. To address this challenge, recent NAS methods shift from conducting architecture search in discrete spaces to continuous spaces using different architecture encoders such as SRM [48], MLP [49], LSTM [14] or GCN [50, 51]. However, what lies in common under these methods is that the architecture representation and search direction are jointly optimized by the supervision signal (e.g., accuracies of the selected architectures), which could bias the architecture representation learning and search direction. [52] emphasizes the importance of studying architecture encodings, and we focus on encoding adjacency matrix-based architectures into low-dimensional embeddings in the continuous space. [53] shows that architectures searched without using labels are competitive to their counterparts searched with labels. Different from their approach which performs pretext tasks using image statistics, we use architecture reconstruction objective to preserve the local structure relationship in the latent space.

## 3 *arch2vec*

In this section, we describe the details of *arch2vec*, followed by two downstream architecture search strategies we use in this work.

### 3.1 Variational Graph Isomorphism Autoencoder

#### 3.1.1 Preliminaries

We restrict our search space to the cell-based architectures. Following the configuration in NAS-Bench-101 [23], each cell is a labeled DAG $\mathcal{G} = (\mathcal{V}, \mathcal{E})$, with $\mathcal{V}$ as a set of $N$ nodes and $\mathcal{E}$ as a set of edges. Each node is associated with a label chosen from a set of $K$ predefined operations. A natural encoding scheme of cell-based neural architectures is an upper triangular adjacency matrix $\mathbf{A} \in \mathbb{R}^{N \times N}$ and an one-hot operation matrix $\mathbf{X} \in \mathbb{R}^{N \times K}$. This discrete encoding is not unique, as permuting the adjacency matrix $\mathbf{A}$ and the operation matrix $\mathbf{X}$ would lead to the same graph, which is known as graph isomorphism [31].

### 3.1.2 Encoder

To learn a continuous representation that is invariant to isomorphic graphs, we leverage Graph Isomorphism Networks (GINs) [30] to encode the graph-structured architectures given its better expressiveness. We augment the adjacency matrix as $\tilde{\mathbf{A}} = \mathbf{A} + \mathbf{A}^T$ to transfer original directed graphs into undirected graphs, allowing bi-directional information flow. Similar to [27], the inference model, *i.e.* the encoding part of the model, is defined as:

$$q(\mathbf{Z}|\mathbf{X}, \tilde{\mathbf{A}}) = \prod_{i=1}^{N} q(\mathbf{z}_i|\mathbf{X}, \tilde{\mathbf{A}}), \text{with } q(\mathbf{z}_i|\mathbf{X}, \tilde{\mathbf{A}}) = \mathcal{N}(\mathbf{z}_i|\boldsymbol{\mu}_i, diag(\boldsymbol{\sigma}_i^2)). \tag{1}$$

We use the $L$-layer GIN to get the node embedding matrix $\mathbf{H}$:

$$\mathbf{H}^{(k)} = \text{MLP}^{(k)}\left(\left(1 + \epsilon^{(k)}\right) \cdot \mathbf{H}^{(k-1)} + \tilde{\mathbf{A}}\mathbf{H}^{(k-1)}\right), k = 1, 2, \ldots, L, \tag{2}$$

where $\mathbf{H}^{(0)} = \mathbf{X}$, $\epsilon$ is a trainable bias, and MLP is a multi-layer perception where each layer is a linear-batchnorm-ReLU triplet. The node embedding matrix $\mathbf{H}^{(L)}$ is then fed into two fully-connected layers to obtain the mean $\boldsymbol{\mu}$ and the variance $\boldsymbol{\sigma}$ of the posterior approximation $q(\mathbf{Z}|\mathbf{X}, \tilde{\mathbf{A}})$ in Eq. (1). During the inference, the architecture representation is derived by summing the representation vectors of all the nodes.

### 3.1.3 Decoder

Our decoder is a generative model aiming at reconstructing $\hat{\mathbf{A}}$ and $\hat{\mathbf{X}}$ from the latent variables $\mathbf{Z}$:

$$p(\hat{\mathbf{A}}|\mathbf{Z}) = \prod_{i=1}^{N}\prod_{j=1}^{N} P(\hat{A}_{ij}|\mathbf{z}_i, \mathbf{z}_j), \text{with } p(\hat{A}_{ij} = 1|\mathbf{z}_i, \mathbf{z}_j) = \sigma(\mathbf{z}_i^T \mathbf{z}_j), \tag{3}$$

$$p(\hat{\mathbf{X}} = [k_1, ..., k_N]^T|\mathbf{Z}) = \prod_{i=1}^{N} P(\hat{\mathbf{X}}_i = k_i|\mathbf{z}_i) = \prod_{i=1}^{N} \text{softmax}(\mathbf{W}_o\mathbf{Z} + \mathbf{b}_o)_{i,k_i}, \tag{4}$$

where $\sigma(\cdot)$ is the sigmoid activation, softmax$(\cdot)$ is the softmax activation applied row-wise, and $k_n \in \{1, 2, ..., K\}$ indicates the operation selected from the predifined set of $K$ opreations at the $\text{n}^{th}$ node. $\mathbf{W}_o$ and $\mathbf{b}_o$ are learnable weights and biases of the decoder.

## 3.2 Training Objective

In practice, our variational graph isomorphism autoencoder consists of a five-layer GIN and a one-layer MLP. The details of the model architecture are described in §4. The dimensionality of the embedding is set to 16. During training, model weights are learned by iteratively maximizing a tractable variational lower bound:

$$\mathcal{L} = \mathbb{E}_{q(\mathbf{Z}|\mathbf{X}, \tilde{\mathbf{A}})}[\log p(\hat{\mathbf{X}}, \hat{\mathbf{A}}|\mathbf{Z})] - \mathcal{D}_{KL}(q(\mathbf{Z}|\mathbf{X}, \tilde{\mathbf{A}})||p(\mathbf{Z})), \tag{5}$$

where $p(\hat{\mathbf{X}}, \hat{\mathbf{A}}|\mathbf{Z}) = p(\hat{\mathbf{A}}|\mathbf{Z})p(\hat{\mathbf{X}}|\mathbf{Z})$ as we assume that the adjacency matrix $\mathbf{A}$ and the operation matrix $\mathbf{X}$ are conditionally independent given the latent variable $\mathbf{Z}$. The second term $\mathcal{D}_{KL}$ on the right hand side of Eq. (5) denotes the Kullback-Leibler divergence [54] which is used to measure the difference between the posterior distribution $q(\cdot)$ and the prior distribution $p(\cdot)$. Here we choose a Gaussian prior $p(\mathbf{Z}) = \prod_i \mathcal{N}(\mathbf{z}_i|0, \mathbf{I})$ due to its simplicity. We use reparameterization trick [37] for training since it can be thought of as injecting noise to the code layer. The random noise injection mechanism has been proved to be effective on the regularization of neural networks [55, 56, 37]. The loss is optimized using mini-batch gradient descent over neural architectures.

## 3.3 Architecture Search Strategies

We use reinforcement learning (RL) and Bayesian optimization (BO) as two representative search algorithms to evaluate *arch2vec* on the downstream architecture search.

### 3.3.1 Reinforcement Learning (RL)

We use REINFORCE [10] as our RL-based search strategy as it has been shown to converge better than more advanced RL methods such as PPO [57] for neural architecture search. For RL, the pre-trained embeddings are passed to the Policy LSTM to sample the action and obtain the next state (valid architecture embedding) using nearest-neighborhood retrieval based on L2 distance to maximize accuracy as reward. We use a single-layer LSTM as the controller and output a 16-dimensional output as the mean vector to the Gaussian policy with a fixed identity covariance matrix. The controller is optimized using Adam optimizer [58] with a learning rate of $1 \times 10^{-2}$. The number of sampled architectures in each episode is set to 16 and the discount factor is set to 0.8. The baseline value is set to 0.95. The maximum estimated wall-clock time for each run is set to $1 \times 10^{6}$ seconds.

### 3.3.2 Bayesian Optimization (BO)

We use DNGO [59] as our BO-based search strategy. We use a one-layer adaptive basis regression network with hidden dimension 128 to model distributions over functions. It serves as an alternative to Gaussian process in order to avoid cubic scaling [60]. We use expected improvement (EI) [61] as the acquisition function which is widely used in NAS [45, 49, 50]. The best function value of EI is set to 0.95. During the search process, the pre-trained embeddings are passed to DNGO to select the top-5 architectures in each round of search, which are then added to the pool. The network is retrained for 100 epochs in the next round using the selected architectures in the updated pool. This process is iterated until the maximum estimated wall-clock time is reached.

## 4   Experimental Results

We validate *arch2vec* on three commonly used NAS search spaces. The details of the hyperparameters we used for searching in each search space are included in Appendix .

**NAS-Bench-101.** NAS-Bench-101 [23] is the first rigorous NAS dataset designed for benchmarking NAS methods. It targets the cell-based search space used in many popular NAS methods [62, 63, 15] and contains $423, 624$ unique neural architectures. Each architecture comes with pre-computed validation and test accuracies on CIFAR-10. The cell consists of 7 nodes and can take on any DAG structure from the input to the output with at most 9 edges, with the first node as input and the last node as output. The intermediate nodes can be either $1 \times 1$ convolution, $3 \times 3$ convolution or $3 \times 3$ max pooling. We split the dataset into 90% training and 10% held-out test sets for *arch2vec* pre-training.

**NAS-Bench-201.** Different from NAS-Bench-101, the cell-based search space in NAS-Bench-201 [24] is represented as a DAG with nodes representing sum of feature maps and edges associated with operation transforms. Each DAG is generated by 4 nodes and 5 associated operations: $1 \times 1$ convolution, $3 \times 3$ convolution, $3 \times 3$ average pooling, skip connection and zero, resulting in a total of $15, 625$ unique neural architectures. The training details for each architecture candidate are provided for three datasets: CIFAR-10, CIFAR-100 and ImageNet-16-120 [64]. We use the same data split as used in NAS-Bench-101.

**DARTS search space**. The DARTS search space [15] is a popular search space for large-scale NAS experiments. The search space consists of two cells: a convolutional cell and a reduction cell, each with six nodes. For each cell, the first two nodes are the outputs from the previous two cells. The next four nodes contain two edges as input, creating a DAG. The network is then constructed by stacking the cells. Following [63], we use the same cell for both normal and reduction cell, allowing roughly $10^{9}$ DAGs without considering graph isomorphism. We randomly sample 600,000 unique architectures in this search space following the mobile setting [15]. We use the same data split as used in NAS-Bench-101.

For pre-training, we use a five-layer Graph Isomorphism Network (GIN) with hidden sizes of {128, 128, 128, 128, 16} as the encoder and a one-layer MLP with a hidden dimension of 16 as the decoder. The adjacency matrix is preprocessed as an undirected graph to allow bi-directional information flow. After forwarding the inputs to the model, the reconstruction error is minimized using Adam optimizer [58] with a learning rate of $1 \times 10^{-3}$. We train the model with batch size 32 and the training loss is able to converge well after 8 epochs on NAS-Bench-101, and 10 epochs on NAS-Bench-201 and DARTS. After training, we extract the architecture embeddings from the encoder for the downstream architecture search.

In the following, we first evaluate the pre-training performance of *arch2vec* (§4.1) and then the neural architecture search performance based on its pre-trained representations (§4.2).

## 4.1 Pre-training Performance

**Observation (1):** We compare *arch2vec* with two popular baselines GAE [27] and VGAE [27] using three metrics suggested by [32]: 1) Reconstruction Accuracy (reconstruction accuracy of the held-out test set), 2) Validity (how often a random sample from the prior distribution can generate a valid architecture), and 3) Uniqueness (unique architectures out of valid generations). As shown in Table 1, *arch2vec* outperforms both GAE and VGAE, and achieves the highest reconstruction accuracy, validity, and uniqueness across all the three search spaces. This is because encoding with GINs outperforms GCNs in reconstruction accuracy due to its better neighbor aggregation scheme; the KL term effectively regularizes the mapping from the discrete space to the continuous latent space, leading to better generative performance measured by validity and uniqueness. Given its superior performance, we stick to *arch2vec* for the remainder of our evaluation.

| Method | NAS-Bench-101 | | | NAS-Bench-201 | | | DARTS | | |
|---|---|---|---|---|---|---|---|---|---|
| | Accuracy | Validity | Uniqueness | Accuracy | Validity | Uniqueness | Accuracy | Validity | Uniqueness |
| GAE [27] | 98.75 | 29.88 | 99.25 | 99.52 | 79.28 | 78.42 | 97.80 | 15.25 | 99.65 |
| VGAE [27] | 97.45 | 41.18 | 99.34 | 98.32 | 79.30 | 88.42 | 96.80 | 25.25 | 99.27 |
| *arch2vec (w.o. KL)* | **100** | 30.31 | 99.20 | **100** | 77.09 | 96.57 | 99.46 | 16.01 | 99.51 |
| *arch2vec* | **100** | **44.97** | **99.69** | **100** | **79.41** | **98.72** | **99.79** | **33.36** | **100** |

Table 1: Reconstruction accuracy, validity, and uniqueness of different GNNs.

**Observation (2):** We compare *arch2vec* with its supervised architecture representation learning counterpart on the predictive performance of the latent representations. This metric measures how well the latent representations can predict the performance of the corresponding architectures. Being able to accurately predict the performance of the architectures based on the latent representations makes it easier to search for the high-performance points in the latent space. Specifically, we train a Gaussian Process model with 250 sampled architectures to predict the performance of the other architectures, and report the predictive performance

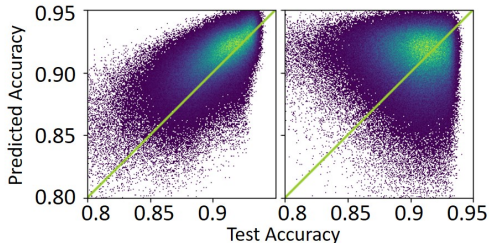

Figure 2: Predictive performance comparison between *arch2vec* (left) and supervised architecture representation learning (right) on NAS-Bench-101.

across 10 different seeds. We use RMSE and the Pearson correlation coefficient (Pearson's r) to evaluate points with test accuracy higher than 0.8. Figure 2 compares the predictive performance between *arch2vec* and its supervised counterpart on NAS-Bench-101. As shown, *arch2vec* outperforms its supervised counterpart[1], indicating *arch2vec* is able to better capture the local structure relationship of the input space and hence is more informative on guiding the downstream search process.

**Observation (3):** In Figure 3, we plot the relationship between the L2 distance in the latent space and the edit distance of the corresponding DAGs between two architectures. As shown, for *arch2vec*, the L2 distance grows monotonically with increasing edit distance. This result indicates that *arch2vec* is able to preserve the closeness between two architectures measured by edit distance, which potentially benefits the effectiveness of the downstream search. In contrast, such closeness is not well captured by supervised architecture representation learning.

**Observation (4):** In Figure 4, we visualize the latent spaces of NAS-Bench-101 learned by *arch2vec* (left) and its supervised counterpart (right) in the 2-dimensional space generated using t-SNE. We overlaid the original colorscale with red (>92% accuracy) and black (<82% accuracy) for highlighting purpose. As shown, for *arch2vec*, the architecture embeddings span the whole latent space, and architectures with similar accuracies are clustered together. Conducting architecture search on such smooth performance surface is much easier and is hence more efficient. In contrast, for the supervised counterpart, the embeddings are discontinuous in the latent space, and the transition of accuracy is non-smooth. This indicates that joint optimization guided by accuracy cannot injectively encode

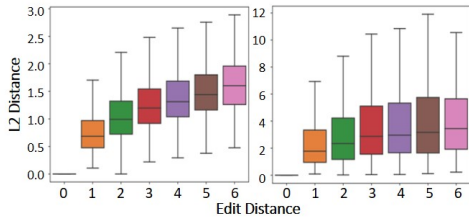 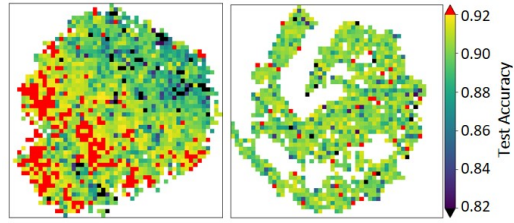

Figure 3: Comparing distribution of L2 distance between architecture pairs by edit distance on NAS-Bench-101, measured by 1,000 architectures sampled in a long random walk with 1 edit distance apart from consecutive samples. left: *arch2vec*. right: supervised architecture representation learning.

Figure 4: Latent space 2D visualization [65] comparison between *arch2vec* (left) and supervised architecture representation learning (right) on NAS-Bench-101. Color encodes test accuracy. We randomly sample 10, 000 points and average the accuracy in each small area.

architecture structures. As a result, architecture does not have its unique embedding in the latent space, which makes the task of architecture search more challenging.

**Observation (5):** To provide a closer look at the learned latent space, Figure 5 visualizes the architecture cells decoded from the latent space of *arch2vec* (upper) and supervised architecture representation learning (lower). For *arch2vec*, the adjacent architectures change smoothly and embrace similar connections and operations. This indicates that unsupervised architecture representation learning helps model a smoothly-changing structure surface. As we show in the next section, such smoothness greatly helps the downstream search since architectures with similar performance tend to locate near each other in the latent space instead of locating randomly. In contrast, the supervised counterpart does not group similar connections and operations well and has much higher edit distances between adjacent architectures. This biases the search direction since dependencies between architecture structures are not well captured.

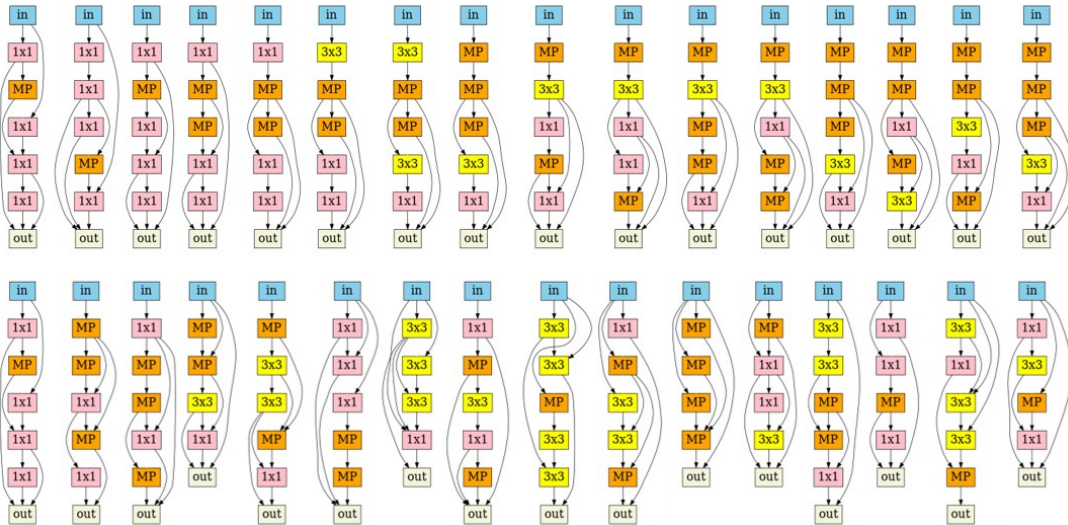

Figure 5: Visualization of a sequence of architecture cells decoded from the learned latent space of *arch2vec* (upper) and supervised architecture representation learning (lower) on NAS-Bench-101. The two sequences start from the same architecture. For both sequences, each architecture is the closest point of the previous one in the latent space excluding previously visited ones. Edit distances between adjacent architectures of the upper sequence are 4, 6, 1, 5, 1, 1, 1, 5, 2, 3, 2, 4, 2, 5, 2, and the average is 2.9. Edit distances between adjacent architectures of the lower sequence are 8, 6, 7, 7, 9, 8, 11, 11, 6, 10, 10, 11, 10, 11, 9, and the average is 8.9.

| NAS Methods | #Queries | Test Accuracy (%) | Encoding | Search Method |
|---|---|---|---|---|
| Random Search [23] | 1000 | 93.54 | Discrete | Random |
| RL [23] | 1000 | 93.58 | Discrete | REINFORCE |
| BO [23] | 1000 | 93.72 | Discrete | Bayesian Optimization |
| RE [23] | 1000 | 93.72 | Discrete | Evolution |
| NAO [14] | 1000 | 93.74 | Supervised | Gradient Decent |
| BANANAS [49] | 500 | 94.08 | Supervised | Bayesian Optimization |
| RL (ours) | 400 | 93.74 | Supervised | REINFORCE |
| BO (ours) | 400 | 93.79 | Supervised | Bayesian Optimization |
| *arch2vec*-RL | **400** | **94.10** | Unsupervised | REINFORCE |
| *arch2vec*-BO | 400 | 94.05 | Unsupervised | Bayesian Optimization |

Table 2: Comparison of NAS performance between *arch2vec* and SOTA methods on NAS-Bench-101. It reports the mean performance of 500 independent runs given the number of queried architectures.

## 4.2 Neural Architecture Search (NAS) Performance

**NAS results on NAS-Bench-101.** For fair comparison, we reproduced the NAS methods which use the adjacency matrix-based encoding in [23][2], including Random Search (RS) [66], Regularized Evolution (RE) [44], REINFORCE [10] and BOHB [12]. For supervised architecture representation learning-based methods, the hyperparameters are the same as *arch2vec*, except that the architecture representation learning and search are jointly optimized. Figure 6 and Table 2 summarize our results.

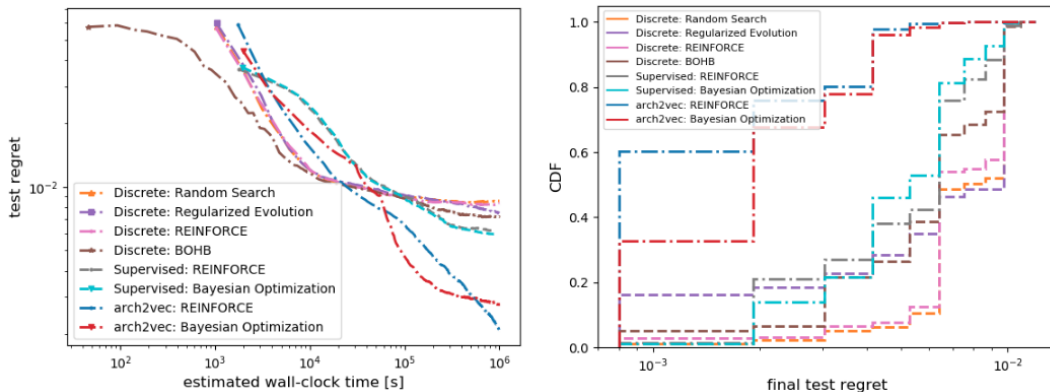

Figure 6: Comparison of NAS performance between discrete encoding, supervised architecture representation learning, and *arch2vec* on NAS-Bench-101. The plot shows the mean test regret (left) and the empirical cumulative distribution of the final test regret (right) of 500 independent runs given a wall-clock time budget of $1 \times 10^6$ seconds.

As shown in Figure 6, BOHB and RE are the two best-performing methods using the adjacency matrix-based encoding. However, they perform slightly worse than supervised architecture representation learning because the high-dimensional input may require more observations for the optimization. In contrast, supervised architecture representation learning focuses on low-dimensional continuous optimization and thus makes the search more efficient. As shown in Figure 6 (left), *arch2vec* considerably outperforms its supervised counterpart and the adjacency matrix-based encoding after $5 \times 10^4$ wall clock seconds. Figure 6 (right) further shows that *arch2vec* is able to robustly achieve the lowest final test regret after $1 \times 10^6$ seconds across 500 independent runs.

Table 2 shows the search performance comparison in terms of number of architecture queries. While RL-based search using discrete encoding suffers from the scalability issue, *arch2vec* encodes architectures into a lower dimensional continuous space and is able to achieve competitive RL-based search performance with only a simple one-layer LSTM controller. For NAO [14], its performance is inferior to *arch2vec* as it entangles structure reconstruction and accuracy prediction together, which inevitably biases the architecture representation learning.

| NAS Methods | CIFAR-10 | | CIFAR-100 | | ImageNet-16-120 | |
|---|---|---|---|---|---|---|
| | validation | test | validation | test | validation | test |
| RE [44] | 91.08±0.43 | 93.84±0.43 | 73.02±0.46 | 72.86±0.55 | 45.78±0.56 | 45.63±0.64 |
| RS [66] | 90.94±0.38 | 93.75±0.37 | 72.17±0.64 | 72.05±0.77 | 45.47±0.65 | 45.33±0.79 |
| REINFORCE [10] | 91.03±0.33 | 93.82±0.31 | 72.35±0.63 | 72.13±0.79 | 45.58±0.62 | 45.30±0.86 |
| BOHB [12] | 90.82±0.53 | 93.61±0.52 | 72.59±0.82 | 72.37±0.90 | 45.44±0.70 | 45.26±0.83 |
| *arch2vec*-RL | 91.32±0.42 | 94.12±0.42 | 73.13±0.72 | 73.15±0.78 | 46.22±0.30 | 46.16±0.38 |
| *arch2vec*-BO | **91.41±0.22** | **94.18±0.24** | **73.35±0.32** | **73.37±0.30** | **46.34±0.18** | **46.27±0.37** |

Table 3: The mean and standard deviation of the validation and test accuracy of different algorithms under three datasets in NAS-Bench-201. The results are calculated over 500 independent runs.

| NAS Methods | Test Error | | Params (M) | Search Cost | | | Encoding | Search Method |
|---|---|---|---|---|---|---|---|---|
| | Avg | Best | | Stage 1 | Stage 2 | Total | | |
| Random Search [15] | 3.29±0.15 | - | 3.2 | - | - | 4 | - | Random |
| ENAS [68] | - | 2.89 | 4.6 | 0.5 | - | - | Supervised | REINFORCE |
| ASHA [69] | 3.03±0.13 | 2.85 | 2.2 | - | - | 9 | - | Random |
| RS WS [69] | 2.85±0.08 | 2.71 | 4.3 | 2.7 | 6 | 8.7 | - | Random |
| SNAS [16] | 2.85±0.02 | - | 2.8 | 1.5 | - | - | Supervised | GD |
| DARTS [15] | 2.76±0.09 | - | 3.3 | 4 | 1 | 5 | Supervised | GD |
| BANANAS [49] | 2.64 | 2.57 | 3.6 | 100 (queries) | - | 11.8 | Supervised | BO |
| Random Search (ours) | 3.1±0.18 | 2.71 | 3.2 | - | - | 4 | - | Random |
| DARTS (ours) | 2.71±0.08 | 2.63 | 3.3 | 4 | 1.2 | 5.2 | Supervised | GD |
| BANANAS (ours) | 2.67±0.07 | 2.61 | 3.6 | 100 (queries) | 1.3 | 11.5 | Supervised | BO |
| *arch2vec*-RL | 2.65±0.05 | 2.60 | 3.3 | 100 (queries) | 1.2 | 9.5 | Unsupervised | REINFORCE |
| *arch2vec*-BO | **2.56±0.05** | **2.48** | 3.6 | 100 (queries) | 1.3 | 10.5 | Unsupervised | BO |

Table 4: Comparison with state-of-the-art cell-based NAS methods on DARTS search space using CIFAR-10. The test error is averaged over 5 seeds. Stage 1 shows the GPU days (or number of queries) for model search and Stage 2 shows the GPU days for model evaluation.

**NAS results on NAS-Bench-201.** For CIFAR-10, we follow the same implementation established in NAS-Bench-201 by searching based on the validation accuracy obtained after 12 training epochs with converged learning rate scheduling. The search budget is set to $1.2 \times 10^4$ seconds. The NAS experiments on CIFAR-100 and ImageNet-16-120 are conducted with a budget that corresponds to the same number of queries used in CIFAR-10. As listed in Table 3, searching with *arch2vec* leads to better validation and test accuracy as well as reduced variability among different runs on all datasets.

**NAS results on DARTS search space.** Similar to [49], we set the budget to 100 queries in this search space. In each query, a sampled architecture is trained for 50 epochs and the average validation error of the last 5 epochs is computed. To ensure fair comparison with the same hyparameters setup, we re-trained the architectures from works that *exactly*[3] use DARTS search space and report the final architecture. As shown in Table 4, *arch2vec* generally leads to competitive search performance among different cell-based NAS methods with comparable model parameters. The best performed cells and transfer learning results on ImageNet [67] are included in Appendix.

# 5   Conclusion

*arch2vec* is a simple yet effective neural architecture search method based on unsupervised architecture representation learning. By learning architecture representations without using their accuracies, it constructs a more smoothly-changing architecture performance surface in the latent space compared to its supervised architecture representation learning counterpart. We have demonstrated its effectiveness on benefiting different downstream search strategies in three NAS search spaces. We suggest that it is desirable to take a closer look at architecture representation learning for neural architecture search. It is also possible that designing neural architecture search method using *arch2vec* with a better search strategy in the continuous space will produce better results.

# 6   Acknowledgement

We would like to thank the anonymous reviewers for their helpful comments. This work was partially supported by NSF Awards CNS-1617627, CNS-1814551, and PFI:BIC-1632051.

## Broader Impact

In this paper, we challenge the common practice in neural architecture search and ask the question: does unsupervised architecture representation learning help neural architecture search? We approach this question through two sets of experiments: 1) the predictive performance comparison and 2) the neural architecture search efficiency and robustness comparison of the learned architecture representations using supervised and unsupervised learning. In both experiments, we found unsupervised architecture representation learning performs reasonably well. Current NAS methods are typically restricted to some small search blocks such as Inception cell or ResNet block, and most of them perform equally well with enough human expertise under this setup. With the drastically increased computational power, the design of the search space will be more complex [70] and therefore hugely increases the search complexity. In such case, unsupervised architecture representation learning may benefit many downstream applications where the search space contains billions of network architectures, with only a few of them trained with annotated data to obtain the accuracy. Supervised optimization in such large search spaces might be less effective . In the future, we suggest more work to be done to investigate unsupervised neural architecture search with different meaningful pretext tasks on larger search spaces. A better pre-training strategy for neural architectures leveraging graph neural networks seems to be a promising direction, as the unsupervised learning method introduced in our paper has already shown its simplicity and effectiveness.

## Footnotes

[1]The RMSE and Pearson's r are: 0.038±0.025 / 0.53±0.09 for the supervised architecture representation learning, and 0.018±0.001 / 0.67±0.02 for *arch2vec*. A smaller RMSE and a larger Pearson's r indicates a better predictive performance.

[2]https://github.com/automl/nas_benchmarks

[3]`https://github.com/quark0/darts/blob/master/cnn/train.py`

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
