[Supplementary Material]

# Supplementary Material:
# Does Unsupervised Architecture Representation Learning Help Neural Architecture Search?

**Shen Yan, Yu Zheng, Wei Ao, Xiao Zeng, Mi Zhang**
Michigan State University
{yanshen6,zhengy30,aowei,zengxia6,mizhang}@msu.edu

## A   Pre-training and search details on each search space

As described in §3, we use adjacency matrix and operation matrix as inputs to our neural architecture encoder (§3.1). In this section, we present the search details for NAS-Bench-101 [1], NAS-Bench-201 [2] and DARTS [3] search spaces.

### A.1   NAS-Bench-101

We followed the encoding scheme in NAS-Bench-101 [1]. Specifically, a cell in NAS-Bench-101 is represented as a directed acyclic graph (DAG) where nodes represent operations and edges represent data flow. A $7 \times 7$ upper-triangular binary matrix is used to encode edges. A $7 \times 5$ operation matrix is used to encode operations, input, and output, with the order as {input, $1 \times 1$ conv, $3 \times 3$ conv, $3 \times 3$ max-pool (MP), output}. For cells with less than 7 nodes, their adjacency and operator matrices are padded with trailing zeros. Figure 1 shows an example of a 7-node cell in NAS-Bench-101 search space and its corresponding adjacency and operation matrices.

For RL-based search, we use REINFORCE [4] as the search strategy. We use a one-layer LSTM with hidden dimension 128 as the controller and output a 16-dimensional output as the mean vector to the Gaussian policy with a fixed identity covariance matrix. The controller is optimized using Adam optimizer [5] with learning rate $1 \times 10^{-2}$. The number of sampled architectures in each episode is set to 16 and the discount factor is set to 0.8. The baseline value is set to 0.95. The maximum estimated wall-clock time for each run is set to $1 \times 10^{6}$ seconds.

For BO-based search, we use DNGO [6] as the search strategy. We use a one-layer fully connected network with hidden dimension 128 to perform adaptive basis function regression. We randomly sample 16 architectures at the beginning, and select the top 5 best-performing architectures and then add them to the architecture pool in each architecture sampling iteration. The network is optimized using selected architecture samples in the pool using Adam optimizer with learning rate $1 \times 10^{-2}$ and trained for 100 epochs in each architecture sampling iteration. The best function value of expected improvement (EI) is set to 0.95. We use the same time budget used in RL-based search.

### A.2   NAS-Bench-201

Different from NAS-Bench-101, NAS-Bench-201 [2] employs a fixed cell-based DAG representation of neural architectures, where nodes represent the sum of feature maps and edges are associated with operations that transform the feature maps from the source node to the destination node. To represent the architectures in NAS-Bench-201 with discrete encoding that is compatible with our neural architecture encoder, we first transform the original DAG in NAS-Bench-201 into a DAG with nodes representing operations and edges representing data flow as the ones in NAS-Bench-101. We then use the same discrete encoding scheme in NAS-Bench-101 to encode each cell into an adjacency

Figure 1: An example of the cell encoding in NAS-Bench-101 search space. The left panel shows the DAG of a 7-node cell. The top-right and bottom-right panels show its corresponding adjacency matrix and operation matrix respectively.

matrix and operation matrix. An example is shown in Figure 2. The hyperparameters we used for pre-training on NAS-Bench-201 are the same as described in §4.

For RL-based search, the search is stopped when it reaches the time budget $1.2 \times 10^4$, $5 \times 10^5$, $1.4 \times 10^6$ seconds for CIFAR-10, CIFAR-100, and ImageNet-16-200, respectively. For CIFAR-10, we follow the same implementation established in NAS-Bench-201 by searching based on the validation accuracy obtained after 12 training epochs with converged learning rate scheduling. The discount factor and the baseline value is set to 0.4. All the other hyperparameters are the same as described in §A.1.

For BO-based search, we initially sample 16 architectures and select the best-performing architecture to the pool in each iteration. The best function value of EI is set to 1.0 for all datasets. We use the same search budget as used in RL-based search. All the other hyperparameters are the same as described in §A.1.

Figure 2: An example of the cell encoding in NAS-Bench-201 search space. The top-left and top-right panels show the original and transformed representations of a cell. The bottom-left and bottom-right panels show its corresponding adjacency matrix and operation matrix respectively.

## A.3 DARTS Search Space

The cell in the DARTS search space has the following property: two input nodes are from the output of two previous cells; each intermediate node is connected by two predecessors, with each connection associated with one operation; the output node is the concatenation of all of the intermediate nodes within the cell [3].

Based on these properties, a $11 \times 11$ upper-triangular binary matrix is used to encode edges and a $11 \times 11$ operation matrix is used to encode operations, with the order as $\{c_{k-2}, c_{k-1},$ zero, 3 $\times$ 3 max-pool, 3 $\times$ 3 average-pool, identity, 3 $\times$ 3 separable conv, 5 $\times$ 5 separable conv, 3 $\times$ 3 dilated conv, 5 $\times$ 5 dilated conv, $c_k\}$. An example is shown in Figure 3. Following [7], we use the same cell for both normal and reduction cell, allowing roughly $10^9$ DAGs without considering graph isomorphism. We randomly sample 600,000 unique architectures in this search space following the mobile setting [3]. The hyperparameters we used for pre-training on DARTS search space are the same as described in §4.

We set the computational budget to 100 architecture queries in this search space. In each query, a sampled architecture is trained for 50 epochs and the average validation accuracy of the last 5 epochs is computed. All the other hyperparamers we used for RL-based search and BO-based search are the same as described in §A.1.

Figure 3: An example of the cell encoding in DARTS search space. The top panel shows the cell. The bottom-left and bottom-right panels show its corresponding adjacency matrix and operation matrix respectively.

# B  More details on pre-training evaluation metrics

We split the the dataset into 90% training and 10% held-out test sets for *arch2vec* pre-training on each search space. In §4.1, we evaluate the pre-training performance of *arch2vec* using three metrics suggested by [8]: 1) Reconstruction Accuracy (reconstruction accuracy of the held-out test set) which measures how well the embeddings can errorlessly remap to the original structures; 2) Validity (how often a random sample from the prior distribution can generate a valid architecture) which measures the generative ability the model; and 3) Uniqueness (unique architectures out of valid generations) which measures the smoothness and diversity of the generated samples.

To compute Reconstruction Accuracy, we report the proportion of the decoded neural architectures of the held-out test set that are identical to the inputs. To compute Validity, we randomly pick 10,000 points $\mathbf{z}$ generated by the Gaussian prior $p(\mathbf{Z}) = \prod_i \mathcal{N}(\mathbf{z}_i|0, \mathbf{I})$ and then apply $\mathbf{z} = \mathbf{z} \odot \text{std}(\mathbf{Z}_{train})$ + mean($\mathbf{Z}_{train}$), where $\mathbf{Z}_{train}$ are the encoded means of the training data. It scales the sampled points and shifts them to the center of the embeddings of the training set. We report the proportion of the decoded architectures that are valid in the search space. To compute Uniqueness, we report the proportion of the unique architectures out of valid decoded architectures.

The validity check criteria varies across different search spaces. For NAS-Bench-101 and NAS-Bench-201, we use the NAS-Bench-101[1] and NAS-Bench-201[2] official APIs to verify whether a decoded

| NAS Methods | Params (M) | Mult-Adds (M) | Top-1 Test Error (%) | Comparable Search Space |
|---|---|---|---|---|
| NASNet-A [11] | 5.3 | 564 | 26.0 | Y |
| AmoebaNet-A [12] | 5.1 | 555 | 25.5 | Y |
| PNAS [12] | 5.1 | 588 | 25.8 | Y |
| SNAS [14] | 4.3 | 522 | 27.3 | Y |
| DARTS [3] | 4.7 | 574 | 26.7 | Y |
| *arch2vec*-RL | 4.8 | 533 | 25.8 | Y |
| *arch2vec*-BO | 5.2 | 580 | 25.5 | Y |

Table 1: Transfer learning results on ImageNet.

(a) *arch2vec*-RL        (b) *arch2vec*-BO

Figure 4: Best cell found by *arch2vec* using (a) RL-based and (b) BO-based search strategy.

architecture is valid or not in the search space. For DARTS search space, a decoded architecture has to pass the following validity checks: 1) the first two nodes must be the input nodes $c_{k-2}$ and $c_{k-1}$; 2) the last node must be the output node $c_k$; 3) except the two input nodes, there are no nodes which do not have any predecessor; 4) except the output node, there are no nodes which do not have any successor; 5) each intermediate node must contain two edges from the previous nodes; and 6) it has to be an upper-triangular binary matrix (representing a DAG).

## C  Best found cells and transfer learning results on ImageNet

Figure 4 shows the best cell found by *arch2vec* using RL-based and BO-based search strategy. As observed in [9], the shapes of normalized empirical distribution functions (EDFs) for NAS design spaces on ImagetNet [10] match their CIFAR-10 counterparts. This suggests that NAS design spaces developed on CIFAR-10 are transferable to ImageNet [9]. Therefore, we evaluate the performance of the best cell found on CIFAR-10 using *arch2vec* for ImageNet. In order to compare in a fair manner, we consider the mobile setting [11, 12, 3] where the number of multiply-add operations of the model is restricted to be less than 600M. We follow [13] to use the exactly same training hyperparameters used in the DARTS paper [3]. Table 1 shows the transfer learning results on ImageNet. With comparable computational complexity, *arch2vec*-RL and *arch2vec*-BO outperform DARTS [3] and SNAS [14] methods in the DARTS search space, and is competitive among all cell-based NAS methods under this setting.

## D  More visualization results of each search space

**NAS-Bench-101**. In Figure 5, we visualize three randomly selected pairs of sequences of architecture cells decoded from the learned latent space of *arch2vec* (upper) and supervised architecture representation learning (lower) on NAS-Bench-101. Each pair starts from the same point, and each architecture is the closest point of the previous one in the latent space excluding previously visited ones. As shown, architecture representations learned by *arch2vec* can better capture topology and operation similarity than its supervised architecture representation learning counterpart. In particular,

Figure 5 (a) and (b) show that *arch2vec* is able to better cluster straight networks, while supervised learning encodes straight networks and networks with skip connections together in the latent space.

**NAS-Bench-201**. Similarly, Figure 6 shows the visualization of five randomly selected pairs of sequences of decoded architecture cells using *arch2vec* (upper) and supervised architecture representation learning (lower) on NAS-Bench-201. The red mark denotes the change of operations between consecutive samples. Note that the edge flow in NAS-Bench-201 is fixed; only the operator associated with each edge can be changed. As shown, *arch2vec* leads to a smoother local change of operations than its supervised architecture representation learning counterpart.

**DARTS Search Space**. For the DARTS search space, we can only visualize the decoded architecture cells using *arch2vec* since there is no architecture accuracy recorded in this large-scale search space. Figure 7 shows an example of the sequence of decoded neural architecture cells using *arch2vec*. As shown, the edge connections of each cell remain unchanged in the decoded sequence, and the operation associated with each edge is gradually changed. This indicates that *arch2vec* preserves the local structural similarity of neighborhoods in the latent space.

(a) *arch2vec* (upper) and supervised architecture representation learning (lower).

(b) *arch2vec* (upper) and supervised architecture representation learning (lower).

(c) *arch2vec* (upper) and supervised architecture representation learning (lower).

Figure 5: Visualization of decoded neural architecture cells on NAS-Bench-101.

(a) *arch2vec* (upper) and supervised architecture representation learning (lower).

(b) *arch2vec* (upper) and supervised architecture representation learning (lower).

(c) *arch2vec* (upper) and supervised architecture representation learning (lower).

(d) *arch2vec* (upper) and supervised architecture representation learning (lower).

(e) *arch2vec* (upper) and supervised architecture representation learning (lower).

Figure 6: Visualization of decoded neural architecture cells on NAS-Bench-201.

Figure 7: Visualization of decoded neural architecture cells using *arch2vec* on DARTS search space. It starts from a randomly sampled point. Each architecture in the sequence is the closest point of the previous one in the latent space excluding previously visited ones.

## Footnotes

[1]`https://github.com/google-research/nasbench/blob/master/nasbench/api.py`

[2]`https://github.com/D-X-Y/NAS-Bench-201/blob/v1.1/nas_201_api/api.py`