[Reviews · NeurIPS 2020]

Review 1

Summary and Contributions: In this paper, the authors have proposed a simple yet effective unsupervised architecture representation learning method for neural architecture search. They have demonstrated that pre-training architecture representations help to build a smoother latent space w.r.t architecture performance. In addition, the pre-trained architecture representations considerably benefit the downstream architecture search.

Strengths: (1)This paper is well written and easy to follow. The authors have clearly presented their approach. (2) The motivation of this paper is easy to understand. (3) The authors have provided clear implementation of their approach. Also, they have provided the code to re-implement their approach. The reviewer has checked their code, and re-run their experiments. (4) The authors have conducted extensive experiments to confirm their contributions.

Weaknesses: (1) how the variational graph isomorphism auto-encoder is designed? (2) In the reinforcement learning module, how did the authors define the observation? (3) How did the authors train the searched network architecture? Train from scratch or ?

Correctness: Yes

Clarity: Yes

Relation to Prior Work: Yes, the authors have clarified the differences from previous papers.

Reproducibility: Yes

Additional Feedback:


Review 2

Summary and Contributions: Induces embeddings by training architectures without access to the training accuracies. To this end, the authors rely on a reconstruction objective that noises the adjacency matrix to make it undirected, then encodes it along with the operations into a code vector, and finally tries to recover the original directed adjacency matrix and operations from the code vector. These code vectors are then used to augment architecture search methods (Bayesian optimization and reinforcement learning) in three search spaces (NAS-Bench 101, NAS-Bench-201, and NasNet). The methods that use the pretrained embeddings achieve sligthly better performance than other competitive baselines.

Strengths: The authors have an interesting hypothesis that using pretrained embeddings learned just based on the structure of the networks can help with architecture search. The experimental results are in Section 4.2 are compelling, showing that the proposed algorithms consistently yield improvements to the architecture found across the three search spaces.

Weaknesses: Even despite the experimental exploration in Section 4.1, it is not clear to me what the pretrained embeddings are capturing about the architectures and why they ought to be good for architecture search. There is no supervision coming from performance, so why would the embeddings be useful The performance improvements are modest.

Correctness: The claims and methods appear to be correct.

Clarity: Yes, although a few aspects could be improved (see feedback).

Relation to Prior Work: Yes.

Reproducibility: No

Additional Feedback: Why is does the colorscale on Figure 4 contains a jumps (red and black)? It wasn't clear for me what the authors were trying to convey in this figure (that somehow the colors are better separated in when using reconstruction embeddings than when using supervised embeddings). Observations 1-5 in Section 4.1 and 4.2 could be named more aptly to reflect the nature of the observation in that paragraph. Is the supervised learning approach one that simply directly tries to predict the \hat A and \hat X without going through the variational autoencoder approach? This should be written down more explicitly (e.g., in a sentence). I couldn't find an explict discussion of this part of the model I didn't find it clear how the pretrained embeddings are used with BO and RL for architecture search (e.g., where are these embeddings passed to the model; how are they used for intermediate states of the controller in the RL (is the code vector computed just with the partial adjacency matrix filled so far?). Perhaps this could be explained more explicitly. Would fine tuning the embeddings during search based on the performances of the architectures, improve the results further? --- I've read the author response and updated my score accordingly.


Review 3

Summary and Contributions: This paper and some recent work [1] [2] share similar motivation that the way each architecture is encoded may have a significant effect on the performance of NAS algorithms. This paper studies the effect of architecture representations for NAS with extensive empirical study. It shows that by incorporating some architecture properties into pre-training (in this work they use structure similarity), networks in the given search space with similar accuracy are able to be clustered together rather than randomly. This will provide a better predictive performance of the latent architecture embeddings, that may further benefit the neural architecture search. [1] Graph Structure of Neural Networks. ICML 2020. [2] A Study on Encodings for Neural Architecture Search, arXiv 2007.04965.

Strengths: This paper shows that optimization on a smoothly-changing performance surface in the latent architecture representation space could lead to more efficient downstream neural architecture sampling process, while joint optimization of neural architecture representation learning and search fails to do so. To construct such space, it pre-trains the architecture representation optimized with structure-level reconstruction loss using standard variational Bayesian methods. In this way, the graph structure of the network is embedded in the latent space given the assumption that similar network structures may have similar accuracies. Figure 4 in the paper validates this assumption, where in most cases by clustering similar network structures, networks with similar performance are grouped together rather than randomly, and modelling on such latent space would be easier for different downstream architecture search methods compared to joint optimization of architecture representations and search. The NAS result using two representative search methods on different search spaces is aligned with its main claim.

Weaknesses: Similar to this work, many recent NAS approaches perform architecture search in the continuous representation space, and gradient descent (GD) is one of the most commonly used approaches for architecture search in the continuous space. However, this work only considers RL and BO as the search algorithms. I am curious to know how competitive GD is compared to RL and BO. In Figure 4, I am not sure why the right plot has more blank space compared to the left plot. An explanation is needed to help the readers understand the insights behind those plots. There are many different ways to describe the similarity between two neural networks. While this work focuses on structural similarity, I would suggest the authors to take a look at the concurrent work [1, 2] which focuses on graph relation similarity and computation similarity respectively, and include a discussion on them. Lastly, this work adopts a similar approach as many current NAS methods where the search space is based on a cell or a ResNet block with relatively small number of nodes. Therefore, you choose to reconstruct the cell/block as the learning objective. This is technically reasonable but somehow simplifies the learning task. I would suggest a recent progress on unsupervised learning [3] in NLP domain. [3] ELECTRA: Pre-training Text Encoders as Discriminators Rather Than Generators, ICLR 2020

Correctness: Yes.

Clarity: Yes.

Relation to Prior Work: This work shares similar motivations with [1, 2] and a concurrent unsupervised NAS approach [4]. The paper provided a short discussion with [4] but lacks a discussion on [1] [2]. [4] Are Labels Necessary for Neural Architecture Search? ECCV 2020.

Reproducibility: Yes

Additional Feedback: Add extra experiments using GD and compare it with RL and BO. Provide a better explanation of Figure 4 (right). Add a discussion on how this work differs from [1] [2].


Review 4

Summary and Contributions: The author propose a graph VAE framework to do NAS tasks. The author argues that the proposed approach is scalable and benefits the architecture sampling approach. The author conduct a series of experiments to support the statement.

Strengths: The proposed idea is straight forward and reasonable. The graph VAE models have been validated to be effective in many graph-related tasks. The experiments are performed in a good quality . Both the qualitative and quantitive results are looking significant and convincing.

Weaknesses: My major concern is the novelty of this work. Although the graph VAE may not been used in NAS before, however, the similar ideas have been proposed in other graph tasks, e.g. https://arxiv.org/pdf/1802.03480.pdf and https://grlearning.github.io/papers/118.pdf . Also, the effectiveness of KL is not fully justified.

Correctness: yes

Clarity: yes

Relation to Prior Work: yes

Reproducibility: Yes

Additional Feedback:

[Author Response · NeurIPS 2020]

We thank all the reviewers for their insightful comments! All the responses will be incorporated into our revision.

**R1**: **(1)** We designed a variational graph isomorphism network to injectively encode structural information of networks in the latent space and accurately remap to original structures after latent space optimization. **(2)** The observations are pretrained embeddings of the selected neural architectures. **(3)** The searched networks are trained from scratch.

**R2**: **(1)** Details of supervised learning approach: architecture embeddings and search strategies (e.g., BO) are jointly optimized in a supervised manner. The supervision signal for embedding learning comes from the accuracies of architectures selected by the search strategies. In addition to accuracy, NAO takes the reconstruction loss of $\hat{A}$ and $\hat{X}$ into account. However, as reported in our submission or Table 1 below, its performance is inferior to our unsupervised approach as it cannot necessarily improve embedding learning due to entangling structure reconstruction and accuracy prediction together. **(2)** Superiority of pretrained embeddings: compared to supervised embeddings, the pretrained embeddings are able to better capture the structural information (e.g. edit distance measures) of neural networks. This is because the optimization objective in pretraining is structure reconstruction only. As we showed in Figure 3 and 4 in the submission, compared to supervised learning, pretraining makes similar architectures clustered better (Figure 3), and hence the accuracies are clustered and distributed more smoothly in the latent space (Figure 4). Conducting architecture search in such smooth performance surface is much easier and is hence more efficient. Note that we only use the accuracy of architecture as supervision in the search phase. **(3)** How pretrained embeddings are used with BO and RL for architecture search: for BO, the pretrained embeddings are passed to Bayesian optimization algorithm (DNGO) to select the top-K architectures in each round of search. For RL, the pretrained embeddings are passed to the Policy LSTM to sample the action and obtain the next state (valid architecture embedding) using nearest-neighborhood retrieval to maximize accuracy as reward. We covered some details in Supplementary A. We will add a thorough description of how pretrained embeddings are used with search strategies in the revision. **(4)** Fine-tuning: we did not fine-tune the embeddings during search based on the performance of the architectures. This is also because it biases the structural clustering obtained from pretraining, which leads to inferior search performance. We will add this result in the revised version. **(5)** Colorscale jumps (red and black) in Figure 4: we overlaid the original colorscale with red (>92% accuracy) and black (<82% accuracy) for highlighting purpose. **(6)** Naming observations: we will name our observations to reflect their nature in the revision. **(7)** Reproducibility: to facilitate fully reproducing our results, we attached the source code in our submitted supplementary material.

**R3**: **(1)** We report the result of GD on NAS101 in terms of test regret in Figure 1 and number of samples in Table 1. We have two observations. First, for GD, NAS with pretrained embeddings outperforms supervised embeddings. This aligns with our results in RL and BO. Second, GD performs worse than RL and BO in both unsupervised and supervised methods. This could be attributed to how GD minimizes the prediction error, which could easily enter the local minimum. We will add this result in the revised version. **(2)** Supervised embeddings are less capable of preserving the structural information due to the learning bias introduced by predicted accuracy, and thus are distributed less smoothly in the latent space which results in more overlapped (or blank) areas.

Figure 1: Test regret of GD & others.

**(3)** Thanks for suggesting [1,2]. [1] focuses on network generators that output relational graphs, and the predictive performance highly depends on the structure measures of the relational graphs. In contrast, we encode structural information of neural networks into compact continuous embeddings, and the predictive performance depends on how well the structure is injected into the embeddings. [2] focuses on transforming adjacency matrix-based encoding to path-based encoding in the discrete space. In contrast, we focus on encoding adjacency matrix-based architectures to low-dimensional embeddings in the continuous space. We will add the discussions on [1,2] in the revised version.

**R4**: **(1)** Thanks for suggesting the related work. While the related work tackles the generative problems, our work focuses on mapping the finite discrete neural architectures into the continuous latent space regularized by KL-divergence such that each architecture is encoded into a unique area in the latent space. Importantly, we systematically investigate how pretraining preserves the structure of neural networks and affects their predictive performance in NAS. We will emphasize this distinction in the revised version. **(2)** The KL term is used to regularize the mapping from the discrete space to the continuous latent space. It helps to perform a better inference and to preserve the validity performance of the model. We show the effectiveness of using KL for pretraining on three search spaces in Table 2 below. We will add this result in the revision.

| NAS Methods | #Queries | Accuracy (%) | Encoding | Search Method |
|---|---|---|---|---|
| NAO | 1000 | 93.74 | Supervised | GD |
| GD (ours) | 400 | 93.69 | Supervised | GD |
| RL (ours) | 400 | 93.74 | Supervised | REINFORCE |
| BO (ours) | 400 | 93.79 | Supervised | BO |
| *arch2vec*-GD | 400 | 93.85 | Unsupervised | GD |
| *arch2vec*-RL | **400** | **94.10** | Unsupervised | REINFORCE |
| *arch2vec*-BO | 400 | 94.05 | Unsupervised | BO |

Table 1: Number of samples of GD & others.

| Method | NAS-Bench-101 | | | NAS-Bench-201 | | | DARTS | | |
|---|---|---|---|---|---|---|---|---|---|
| | Accuracy | Validity | Uniqueness | Accuracy | Validity | Uniqueness | Accuracy | Validity | Uniqueness |
| *arch2vec (w.o. KL)* | **100** | 30.31 | 99.20 | **100** | 77.09 | 96.57 | 99.46 | 16.01 | 99.51 |
| *arch2vec* | **100** | **51.33** | **99.36** | **100** | **79.41** | **98.72** | **99.79** | **33.36** | **100** |

Table 2: An ablation study on the effectiveness of KL for pretraining.

[Meta-Review · NeurIPS 2020]

In the paper authors created an unsupervised learning method to embeds architectures in latent space and showed through experiments that the representations formed result in improved downstream performance compared to training with supervised objective jointly. The idea is important and the analysis is sound. The paper could be improved by analysing more diverse space of architectures than ResNet like blocks, as well as other suggestions given by the reviewers.